# Exploring person-specific associations of situational motivation and readiness with leisure-time physical activity effort and experience

Kelley Strohacker[1]*, Gorden Sudeck[2,3], Adam H. Ibrahim[1], Richard Keegan[4]

1 Department of Kinesiology, Recreation and Sport Studies, The University of Tennessee, Knoxville, TN, United States of America, 2 Institute of Sport Science, University Tübingen, Tübingen, Germany, 3 Interfacultary Research Institute for Sports and Physical Activity, University Tübingen, Tübingen, Germany, 4 Research Institute for Sport and Exercise, Faculty of Health, University of Canberra, Canberra, Australia

* kstrohac@utk.edu

**Data Availability Statement:** All relevant data are within the manuscript and its Supporting Information files.

## Abstract

Identifying determinants of leisure-time physical activity (LTPA) often relies on population-level (nomothetic) averages, potentially overlooking person-specific (idiographic) associations. This study uses an idiographic perspective to explore how subjective readiness and motives for LTPA relate to volitional effort (duration, intensity) and affective experience (pleasure, displeasure). We also highlight the potential for different interpretations when data are averaged within individuals and assessed using a variable-centered approach. Participants (N = 22, 25±8 years old, 54.5% women) were asked to continue their regular PA patterns for 10 weeks. Ecological momentary assessment procedures allowed participants to provide pre-activity reports (physical, cognitive, emotional readiness and situational motive for activity) and post-activity reports (activity type, duration, perceived exertion, ratings of affective valence). Spearman rank correlation was implemented to interpret within- and between-person associations. Data visualization approaches were used to showcase person-specific differences in associations. Participants provided 519 reports of LTPA (24 ±11 events/person), which displayed between- and within-person variety in type, duration, intensity, and affective experience. Exemplar cases highlight discrepancies in interpretation based on level of analysis, such that the nomothetic association (rho = .42, p = .05; 95% CI -.02, .72) between motive to replenish energy and LTPA duration was observed in only one within-person analysis (41% were weak-to-large inverse effects). Alternatively, the negligible nomothetic association (rho = .02, p = .93; 95% CI -.41, .44) between physical readiness and LTPA-related affect did not reflect the 59% of within-person analyses showing moderate-to-large, positive effects. Future research aiming to identify determinants of LTPA effort and experience should integrate contemporary, idiographic analyses in early-stage research for developing person-specific strategies for LTPA promotion.

**Funding:** The author(s) received no specific funding for this work.

**Competing interests:** The authors have declared that no competing interests exist.

## Introduction

Leisure-time physical activity (LTPA) broadly refers to activities performed in one's free time based on personal interests and needs, which includes formal exercise training as well as activities such as walking, hiking, sport, or dance [1]. Regular engagement in LTPA protects against physical health conditions (cardiovascular disease, type 2 diabetes, renal disease, certain cancers) that serve as leading causes of mortality and disability in Westernized countries [2, 3]. Mental health benefits have also been associated with LTPA [4]. However, insufficient physical activity remains a global public health problem [5]; in the United States, for example, 26% of adults report engaging in no LTPA at all [6], and maintaining consistent patterns of LTPA appears relatively difficult for the majority of adults [7–9].

Strategies for LTPA promotion are informed by identifying behavioral determinants. Determinants of physical activity are numerous and multifaceted, spanning multiple classifications [10, 11]. Relating to individuals, for example, Self-Determination Theory supports that psychological needs (e.g., perceived autonomy, competence, and relatedness) must be met to elicit intrinsic motivation [12]. Additionally, the characteristics of physical activity itself (e.g., intensity, perceived effort) are considered determinants of future behavior [10]. It is critical to note, however, that determinants generally have been inferred based on variable-centered approaches (i.e., averaging across pooled samples, with summary statistics pertaining to the aggregate); however, it cannot be assumed that similar inferences can be made for each individual within a given subpopulation [13].

Thus, it is crucial to also understand determinants of LTPA using person-specific and person-centered analyses [14]. Person-specific approaches (i.e., testing within-person associations for individuals separately) can be implemented to estimate inter-individual differences in associations (magnitude and direction) between hypothesized determinants and LTPA-related outcomes [15, 16]. Analysis of person-specific data may uncover observable subpopulations, for which more generalized inferences can be held based on person-centered analyses [17]. Identifying which determinants of LTPA are unique to a person, are shared within an identifiable subgroup, and are shared across the general population aligns with expert-led calls to view physical activity as a complex and dynamic health behavior, requiring interventions that are individualized and person-adaptive [15, 18, 19].

Ecological momentary assessment (EMA) techniques can be used to understand determinants and consequences of physical activity at the person-specific level by repeatedly collecting data from individuals over time, within their natural environment [20]. Existing research using EMA, however, has been criticized for reducing the richness of data in order to conduct traditional statistical approaches (e.g., linear regression, multilevel modeling; [21]) for generalizing relationships across the sample. Few studies to date have explicitly demonstrated evidence for heterogeneity in within-person associations specific to physical activity [22, 23], which were limited to quantitative characteristics of performed physical activity (e.g., duration, intensity, steps per day) that influence physiological adaptations. Beyond that, experiential factors must be accounted for when assessing person-specific associations, as contemporary models and theories in exercise psychology [24–27] identify affective and evaluative responses arising from physical activity as determinants of future intentions and behavior, in general.

The aim of the current paper is to explore interindividual variability in person-specific associations regarding hypothesized predictors of volitional LTPA effort and affective experiences. We also explore the potential for differing interpretations when data are averaged within individuals and then assessed using a variable-centered approach. Target variables were chosen based on a proposed person-adaptive model for exercise behavior (flexible nonlinear periodization; FNLP) [28]. Under this conceptual model, an individual's *situational motivation* for

physical activity (e.g., for fitness, stress relief, social contact) and activity-related *readiness* (availability of physical, cognitive, emotional resources) are hypothesized to influence volitional effort (intensity, duration), as well as affect-related experiences (e.g., pleasure vs. displeasure). Enactment of FNLP (choosing activities specifically based on acute situational motivation and readiness states) incorporates variety, autonomy-support and flexible goal-setting to support factors related to behavior maintenance, such as enjoyment and self-regulation [29].

Importantly, in light of the need for transparency in kinesiology research–including exercise psychology [30]–we emphasize that the work presented herein was designed to be exploratory. To further enhance transparency, we rely on descriptive approaches and visualization [31] to explore the data at this early stage of inquiry. We anticipate that demonstrating proof-of-concept for heterogeneity in association and potentially conflicting interpretations will stimulate multi-level (person-specific, person-centered, variable centered) research to more thoroughly understand determinants of LTPA performance and experience. This should allow for improved knowledge about which constructs are most meaningful within and across individuals, which can effectively guide the development of person-adaptive intervention strategies that support sustained physical activity.

## Methods

### Study design

Adults in the United States and Germany were recruited to participate in research-related activities for 10 consecutive weeks. While enrolled, participants used their personally-owned smartphones to enact event-contingent ecological momentary assessments (pre- and post-activity) in response to purposeful (performed with intention, awareness) physical activity sessions lasting ten or more minutes. The EMA data were used to computed person-specific and variable-centered associations. All procedures described were approved for each site (The University of Tennessee, Knoxville Institutional Review Board; Ethics Committee of the Faculty of Economics and Social Sciences at University Tübingen) and all participants provided written, informed consent to engage in research-related activities by signing a standardized paper consent form, which was witnessed and also signed by the overseeing research associate.

### Inclusivity in global research

Additional information regarding the ethical, cultural, and scientific considerations specific to inclusivity in global research is included in the S1 Checklist.

### Procedures

**Recruitment, orientation, and enrollment.** Participants were recruited in rolling cohorts in the United States (08/23/2022 to 09/27/2022; Southeastern flagship university) and Germany (11/8/2022 to 01/15/2023; Southwest university). Recruitment was conducted using flyer advertisements, social media, classroom announcements, and snowball sampling. Individuals were eligible to participate if they were at least 18 years old, reported engaging in at least 60 minutes per week of moderate-to-vigorous physical activity, on average, and owned a smartphone capable of accessing the Internet. Exclusion criteria included reported visual or cognitive impairments that would limit survey completion or anticipated planned, extended travel where cellular service was limited. Though confirmatory statistics were not conducted the study, recruitment goals and enrollment duration were guided by repeated measures correlation power curves [32], such that a minimum of 10 individuals, each with 10 within-person

data points (i.e., 1 report of LTPA per week) would be required to detect at least medium ($r = 0.3$) effects with 80% power.

Each orientation session consisted of a study overview, informed consent procedures, completion of baseline surveys, the bookmarking of the recurring web survey link in participants' smartphone Internet browser, and recording of height and weight to compute body mass index (six German participants provided only self-reported metrics). Participants were instructed to continue their usual pattern of physical activity behavior and initiate event-contingent surveys starting the day after the orientation visit and continue reporting for the subsequent 10 weeks of enrollment. All participants were emailed weekly reminders to initiate reports and targeted re-engagement emails were sent to those who did not submit at least one activity report over a two-week period. No incentives or compensation were provided for engaging in research activities.

**Event-contingent ecological momentary (EMA) assessment.**   Surveys were built and disseminated using Qualtrics Research Suite (Provo, UT). Upon opening the survey link, participants entered their study identification number and indicated the report's purpose (e.g. submit a momentary pre-activity report; submit a momentary post activity report; to retrospectively report an activity bout) to guide subsequent display logic. The pre-activity survey first captured the domain of the impending physical activity. The answer triggered survey skip logic to display mode options for each domain–leisure-time (e.g., hiking, weightlifting, jogging), transportation (e.g., walking, biking), domestic (e.g., gardening, moving furniture), occupational (e.g., moving heavy objects, manual labor). Each list included an "other" option to type in an unlisted activity. Participants then rated their situational motive(s) for engaging in LTPA based on the revised Bernese Motive and Goal Inventory in Exercise and Sport (33) and provided 'right now' ratings of physical, cognitive, and emotional readiness and fatigue using a shortened version of the Acute Readiness Monitoring Scale (34). The post-activity survey allowed individuals to indicate total activity duration and to recall and rate their perceived exertion (Catgory-10 Ratio Scale; (35)) and affective valence using the Feeling Scale (36), regarding the activity they had performed. Retrospective surveys included all aforementioned items, plus additional items to indicate the date and time that the reported activity was initiated.

## Instrumentation

**Bernese motive and goal inventory in exercise and sport (BMZI; revised version).**   The revised BMZI is a validated [33] instrument used to assess multidimensional motive profiles. Following the prompt "why do you exercise / why would you exercise", participants were asked to rate 25 items on a 5-point Likert scale (1 = I strongly agree, 5 = I strongly disagree). Items represented seven motive domains: contact, performance/competition, distraction/catharsis, body/appearance, aesthetics, fitness, health. The BMZI, in full, was administered one time within the baseline survey packet. Eleven single items were selected considering their representativeness for the motive domain and their appropriateness for situational assessments, with modified to "right now" phrasing in the pre-activity surveys.

**Acute readiness monitoring scale (ARMS).**   The ARMS is a 32-item scale that was designed to assess situational readiness. Ten items were chosen to represent the five factors (of nine, total) deemed pertinent to volitional physical activity behavior: physical readiness ("I feel physically fit"; "I am physically fresh"), physical fatigue ("I am physically tired"; "I am physically spent"), cognitive readiness ("I am thinking clearly"; "I can focus well"), cognitive fatigue ("I cannot focus today"; "I am mentally tired"), threat-challenge readiness ("I have things under control today"; "I can handle unpleasant feelings"). Items are rated on 7-point Likert

scale (0 = does not apply at all; 6 = fully applies). Prior research has supported the psychometric soundness of items [34, 35]. The ARMS was administered at each event-contingent pre-activity survey.

**Feeling scale (FS).** Recalled affective valence during physical activity was assessed using the FS. The FS is an 11-point scale that assesses core affect regarding pleasure and displeasure [36] and has been used extensively to understand experiences relating to both muscle-strengthening activities [37] and aerobic activities [38]. The FS ranges from -5 (very bad) to 5 (very good) with 0 (neutral) as the midpoint and has been validated in German [39]. Rather than using 'right now' framing (which is likely to capture a rebound effect [40]), instructions were provided to elicit ratings as representative as possible of affective experiences while under exertion ('*Overall, how pleasant or unpleasant did you feel WHILE you were physically active*'), as true in-task measures were not able to be captured in this study. The FS was administered at each event-contingent post-activity survey.

**Category-10 ratio (CR-10) scale for ratings of perceived exertion (RPE).** Perceptions of activity-related heaviness and strain was assessed using the CR-10, a validated single-item instrument [41] where respondents provide a rating on an 11-pt scale (0 = rest; 1 = very, very easy; 2 = easy; 3 = moderate; 4 = somewhat hard; 5 = hard; 7 = very hard; 10 = maximal). Scale points 6, 8, and 9 are intentionally blank. The CR-10 was administered at each event-contingent post-activity survey.

**Behavioral regulation in exercise questionnaire (BREQ)-3.** The BREQ-3 is a validated [42] instrument used to assess behavioral regulations in an exercise context using six subscales: amotivation, external regulation, introjected regulation, identified regulation, integrated regulation, and intrinsic regulation. 24 items representing each domain are scored on a 5-point scale (0 = not true, 5 = very true) and the mean value for each subscale is calculated. The relative autonomy index (RAI) is a score derived from all subscales that describes how self-determined respondents are at that time. The RAI is obtained by applying a weighting to each subscale (amotivation = -3, external = -2, introjected = -1, identified = 1, integrated = 2, intrinsic = 3) and then summing the weighted scores. The BREQ-3 was administered one time within the baseline survey packet.

**Physical activity-related health competence (PAHCO).** To assess PAHCO, respondents answered 13 items, scored on a 5-pt scale (1 = disagree completely, 5 = agree completely) related to four areas of competence: 'control competence for physical training', 'physical activity-related affect regulation', 'physical activity-specific self-control', and 'motivational competence'. These instruments have been found to be valid and reliable [43, 44] and were administered one time within the baseline survey packet.

**Perceived physical fitness scale (PPFS).** The PPFS is a validated [45], 12-item scale that assesses respondents' perceptions of their physical fitness across four domains: cardiorespiratory fitness (5 items), muscular strength (3 items), flexibility (2 items), and body composition (2 items). All items are assessed on a 5-pt Likert scale (1 = strongly disagree, 2 = disagree, 3 = undecided, 4 = agree, 5 = strongly agree). Scores are summed within each domain and across all domains to provide an overall perceived physical fitness scores (range 12 to 60). The PPFS was administered within the baseline survey packet.

**World Health Organization–five well-being index (WHO-5).** The WHO-5 is a validated [46] short, self-reported measure of current subjective well-being. Participants are presented with five statements ("I have felt cheerful and in good spirits", "I have felt calm and relaxed", I have felt active and vigorous", "I woke up feeling refreshed and rested", "my daily life has been filled with things that interest me") and asked to provide a frequency rating pertaining to feelings over the last two weeks. Each statement is rated on a 6-point scale (0 = at no time, 1 = some of the time, 2 = less than half of the time, 3 = more than half of the time, 4 = most of

the time, 5 = all of the time). Raw scores (ranging from 0–25) are multiplied by 4 and then interpreted with 0 representing the worst imaginable well-being and 100 representing the best imaginable well-being. The WHO-5 was administered within the baseline survey packet.

**Demographics.**    Within the baseline survey packet, participants provided their age in years, as well as self-identified their gender identity by choosing one or more designations (agender, woman, gender fluid, man, non-binary, transgender, other, prefer not to answer) and highest level of education achieved (high school diploma, GED or alternative credential, some college, Associate's degree, Bachelor's degree, Master's degree, professional degree) at the time of study enrollment. Items to allow self-identification of race and ethnicity (African American/Black, Arab American, Asian American / Pacific Islander, Indigenous / First Nations / Native American, Latino or Hispanic, White, other, prefer not to answer) were presented to the US cohort only, as questions on race and ethnicity are not common for participants in Germany and discomfort by asking participants these questions should be avoided.

## Data processing

Pre-, post-, and retrospective physical activity databases were merged and organized, such that all single LTPA session reports from a given participant were chronologically listed in subsequent rows to support person-specific analyses of association. Columns included survey metadata (e.g., survey duration, date and time of completion) as well as target study variables. The database was further organized to only include exercise-related and non-exercise-related LTPA (i.e., excluding any sessions designated as domestic or occupational). One report from one participant was excluded, as it pertained to an extreme outlier (11-hour activity session; mean duration of remaining 33 sessions = 47.78±14.53 min). Four participants from analyses due to reporting less than 5 LTPA sessions (minimum engagement considered to be study active). A separate database contained within-person averages for all target variables to conduct variable-centered analyses of association.

## Statistical and visual analyses

All analyses described were conducted using the Statistical Package for Social Sciences (SPSS). Basic descriptive and frequency analyses were conducted to describe the sample and summarize within-person readiness states, behavior, and experiences. Coefficients of variation were computed (person-specific standard deviation divided by person-specific mean) to understand degrees of within-person variability regarding LTPA duration, intensity, and affective valence. Individuals demonstrating the lowest, median, and highest coefficient of variation for each outcome were visualized using standard line graphs.

Each individual's most important motives and goals for LTPA were computed using their baseline BMZI data to further describe the participants based on existing classification approach of typical motive and goal profiles for LTPA [47]. Furthermore, radar plots were constructed for each individual to visually demonstrate within-person variability across all reported LTPA sessions regarding the 5 domains of readiness and 11 types of situational motivation.

To account for measuring most variables using ordinal scales, Spearman rank correlation procedures were used to compute size and direction of association (rho; ρ), as well as 95% confidence intervals to explore how pre-activity ratings of situational motivation and readiness related to subsequent activity duration, perceived exertion, and affective responses at the person-specific level. Of the 1,056 idiographic analyses (22 participants x 16 domains for situational motivation and readiness x 3 LTPA outcomes) there were 48 instances (across 6 participants) where analyses could not be conducted due to the individual reporting no

variance in one correlate. When averaging session-to-session ratings within each person, data were kept in their ordinal form (zero decimal places) and we repeated the Spearman rank correlation processes to explore variable-centered associations. Forest plots were constructed to visualize interindividual variance in person-specific associations, whereas scatterplots of ranked values were constructed to visualize variable-centered associations.

Heat maps were constructed to broadly visualize the proportion of individuals (red = 0 individuals; dark green = highest number of individuals) assigned across seven categories: negligible association; negative associations (small, medium, large); positive associations (small, medium, large) based on idiographic analyses. Each heat map was organized first based on hypothesized determinant (motivation type or readiness domain), then listed in order from the strongest positive to the strongest negative nomothetic association. For the purpose of highlighting potential differences in interpretation, associations are referred to as small (.10 to .29), medium (.30-.49), or large (>.50) based on standard demarcations [48].

## Results

In general, participants retained for analysis (N = 22) were 25±8 years of age (range 18–56 years) with an average body mass index of 22.8±3.6 kg/m$^2$ (range 17.1–33.2 kg/m$^2$). All participants identified as either a man (45.5%) or a woman (54.5%), with four participants reporting having achieved a graduate degree (Master's, Doctorate) at the time of study enrollment. Of the participants retained from the US cohort (n = 7), six self-identified as non-Hispanic White, with one participant identifying as both White and Hispanic. Participants generally presented scores in the upper end of the scale ranges for physical activity-related health competencies, autonomous exercise regulation, and perceived overall physical fitness (Table 1).

Overall, the 22 participants provided a total of 519 reports of LTPA (24±11; min = 8, max = 50) for analysis. Table 2 demonstrates idiographic differences in number of reported LTPA sessions, most important motives and goals, and variety in LTPA types reported.

To further demonstrate the degree of observed inter- and intra-individual variability in LTPA duration, intensity, and affective experience, Fig 1 highlights individuals demonstrating the lowest, median, and highest coefficients of variation regarding LTPA duration (Panel A), intensity (Panel B), and affective valence (Panel C) across all reported sessions. Fig 2 provides two exemplar cases to demonstrate the potential for within-person and between-person differences regarding dynamics of readiness and situational motivation states. One individual

**Table 1. Summary of participant (N = 22) baseline traits.**

| Variable | Mean ± standard deviation |
|---|---|
| PAHCO Control Competence [1–5] | 3.98 ± .55 |
| PAHCO Affect Regulation [1–5] | 4.48 ± .50 |
| PAHCO Self-Control [1–5] | 4.18 ± .66 |
| PAHCO Motivational Competence [1–5] | 4.38 ± .64 |
| Relative Autonomy Index [−19–24] | 19.27 ± 2.49 |
| PPFS Muscular Strength [3–15] | 9.92 ± 2.31 |
| PPFS Flexibility [2–10] | 5.92 ± 2.78 |
| PPFS Body Composition [2–10] | 8.00 ± 1.91 |
| PPFS Cardiorespiratory [5–25] | 18.58 ± 3.00 |
| PPFS Overall Perceived Fitness [12–60] | 42.42 ± 6.42 |
| WHO-5 Well-Being Index [0–100] | 60.55 ± 12.42 |

PAHCO = physical activity-related health competence questionnaire; PPFS = perceived physical fitness scale

**Table 2. Person-specific descriptions of motives and reported modes of leisure-time physical activity (LTPA).**

| Participant ID (# sessions reported) | Most Important Motives for LTPA | Reported Modes (% of reports containing each mode) |
|---|---|---|
| P001 (8) | C/P; D/SR; A | Running (100%) |
| P002 (14) | D/SR; C/P | Weight Lifting (79%); Body Weight Calisthenics (50%); Yoga (43%); 'Cardio' (7%) |
| P005 (33) | D/SR; A; H | Walking (94%); Weight Lifting (6%) |
| P006 (25) | D/SR; C/P | Weight Lifting (52%); Walking (36%); Biking (16%); Body Weight Calisthenics (12%); 'Ergometer' (8%) |
| P007 (16) | F; BW/A; D/SR | Weight Lifting (75%); Sprints (25%); Running (6%) |
| P008 (29) | C/P; F | Biking (69%); Weight Lifting (17%); Body Weight Calisthenics (7%); Walking (3%) |
| P009 (15) | F; A | Weight Lifting (80%); Biking (20%); Running (13%); 'Ergometer' (7%) |
| P010 (31) | D/SR; BW/A; F | Running (97%); Walking (3%) |
| P101 (44) | BW/A; H | Tennis (36%); Roundnet (27%); Running (14%); Skiing (14%); Walking (5%); Hiking (5%); Volleyball (5%) |
| P102 (26) | SC | Body Weight Calisthenics (54%); Roundnet (27%); Running (12%); Weight Lifting (12%) |
| P103 (39) | SC; F | Weight Lifting (62%); Roundnet (26%); Body Weight Calisthenics (10%); Walking (5%); Yoga (3%); 'Fitness' (3%) |
| P104 (13) | F; H | Weight Lifting (69%); Running (8%); Walking (8%); Football (8%); Skiing (8%) |
| P105 (14) | F; BW/A; SC | Roundnet (64%); Running (21%); Floorball (14%) |
| P106 (34) | F; D/SR; H | Body Weight Calisthenics (65%); Weight Lifting (55%); Running (50%); 'Fitness' (10%); 'Workout' (5%); Swimming (5%); Walking (5%) |
| P107 (11) | F; SC; C/P | Badminton (18%); Soccer (18%); Weight Lifting (18%); Running (18%); Bouldering (9%); Body Weight Calisthenics (9%); Skiing (9%) |
| P108 (27) | F; A | Running (67%); Body Weight Calisthenics (15%); Yoga (11%); Bouldering (7%); Hiking (4%) |
| P109 (50) | A | Skiing (28%); Climbing (20%); Walking (14%); Running (10%); 'Stable Work' (8%); Dancing (6%); Body Weight Calisthenics (6%); Yoga (4%); Horseback Riding (6%) |
| P110 (24) | F; H | Weight Lifting (100%) |
| P112 (18) | F; BW/A | Bouldering (56%); Weight Lifting (28%); Body Weight Calisthenics (11%); Running (6%) |
| P010 (31) | D/SR; BW/A; F | Running (97%); Walking (3%) |
| P121 (32) | SC; D/SR; A | Climbing (75%); Body Weight Calisthenics (13%); Running (9%); Walking (3%); Weight Lifting (3%) |
| P122 (15) | D/SR; BW/A; A; F | Bouldering (67%); Climbing (7%); Parkour (7%); Body Weight Calisthenics (7%); Snowboarding (7%); Walking (7%) |
| P123 (15) | D/SR; F; A | Bouldering (53%); Yoga (40%); Weight Lifting (7%); Body Weight Calisthenics (7%); Ballet (7%) |

A = Aesthetics; BW/A = Body Weight/Appearance; C/P = Competition/Performance; D/SR = Distraction/Stress Regulation; F = Fitness; H = Health; SC = Social Contact

(P008) demonstrated more consistency in rating readiness relatively high and fatigue relatively low, with little motivation to perform their 29 LTPA sessions for distraction/stress reduction (e.g., organize thoughts, reduce stress) or for weight regulation / body shape (e.g., regulate weight, shape my body). In comparison, the second individual (P121) reported 32 LTPA

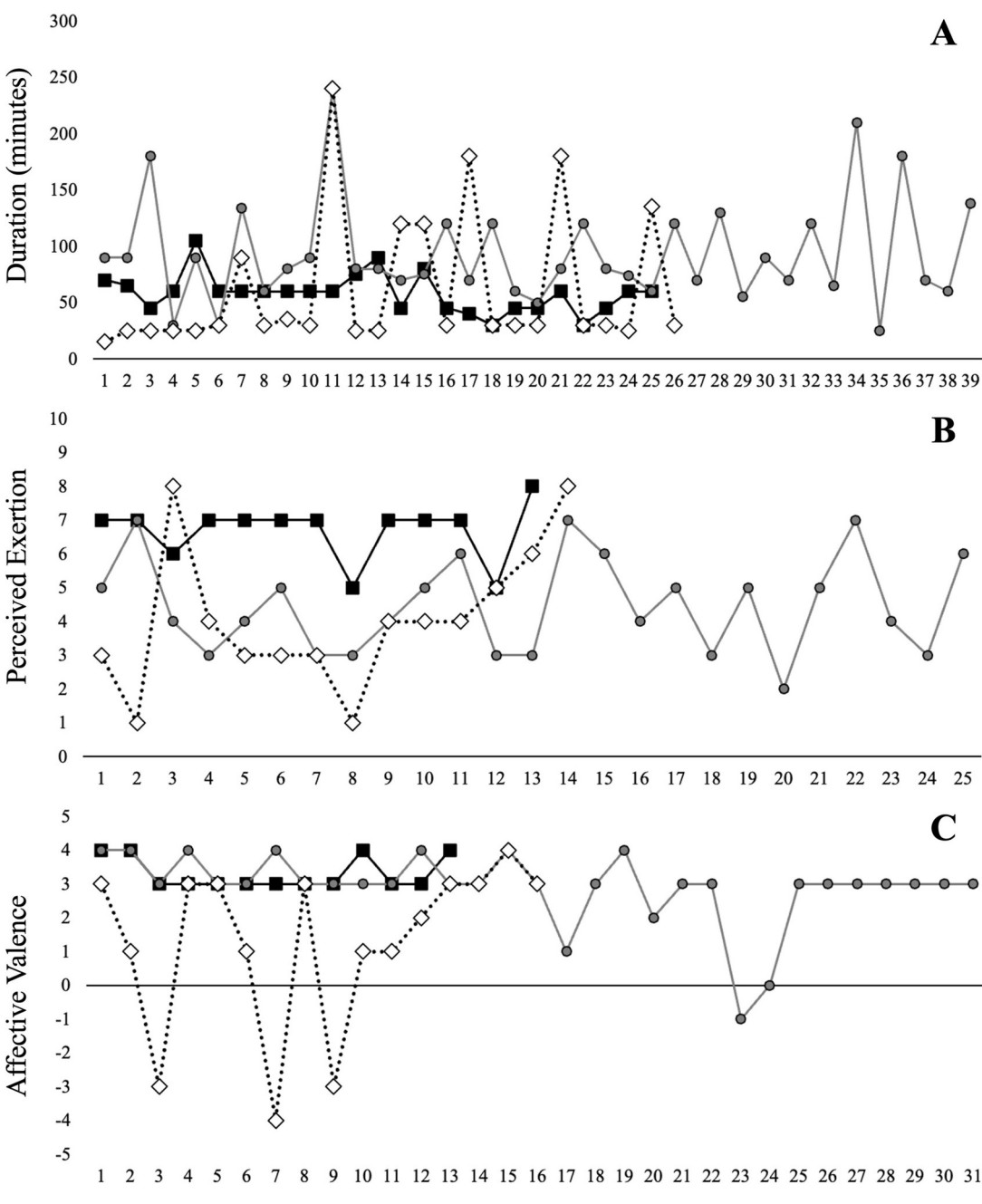

**Fig 1. Individuals' leisure-time physical activity (LTPA) sessions vary over time in duration, intensity, and recalled experienced pleasure.** Participants (N = 22) provided ecological momentary assessments across 10 weeks to indicate total minutes per session (Panel A), overall ratings of perceived exertion (category-10 ratio scale; Panel B), and recalled affective valence (feeling scale; Panel C). Representative participants are those presenting with the lowest (black squares), median (gray circles), and highest (white diamonds) coefficient of variation in each target outcome.

sessions, which were preceded with rather high variation across readiness and fatigue states, as well across most types of situational motivation.

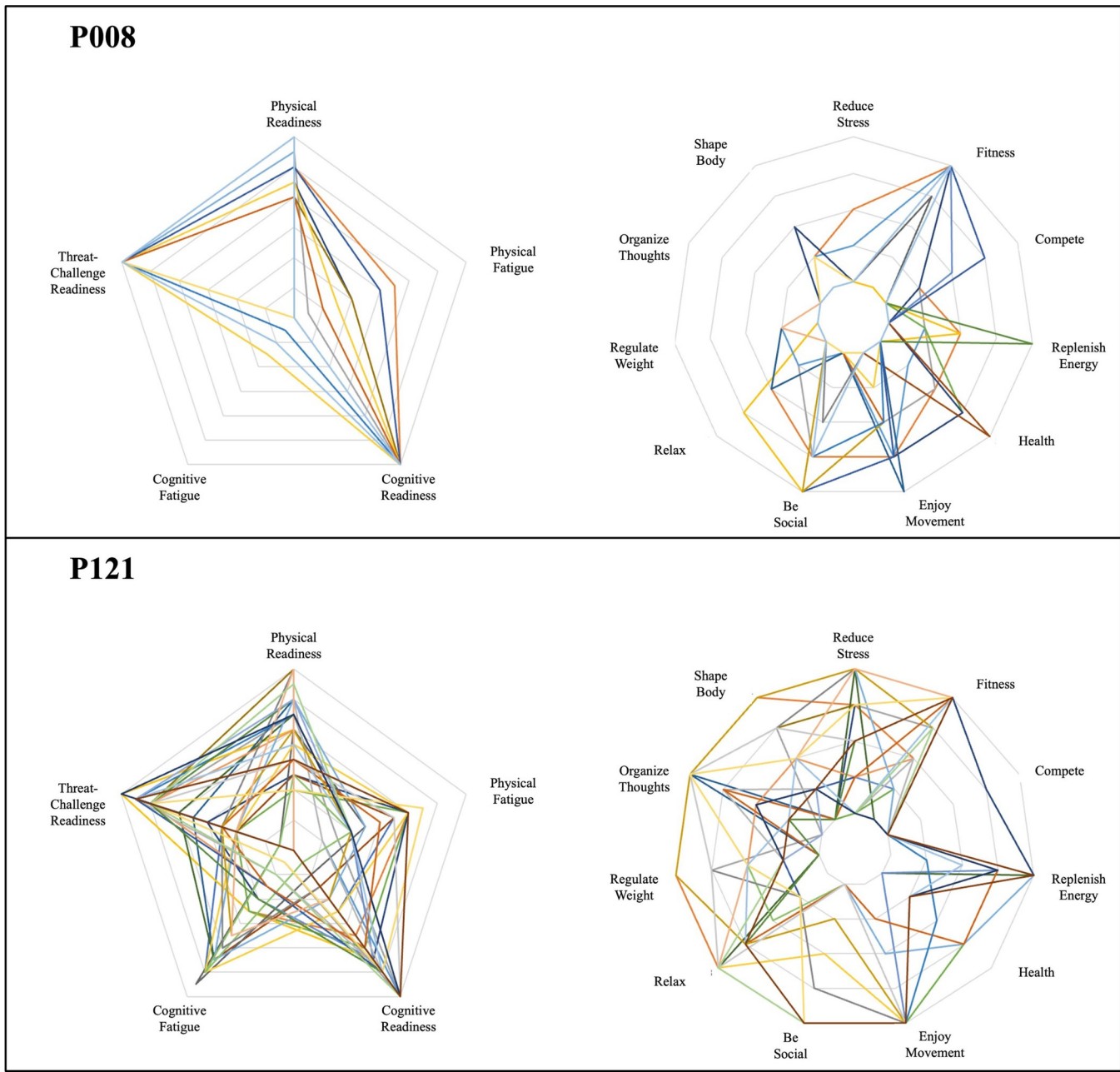

**Fig 2. Pre-activity perceptions of readiness and situational motivation vary across sessions, with differing degrees of variance across individuals.**
Participants (N = 22) provided ecological momentary assessments across 10 weeks to indicate readiness states and motives to prior to performing leisure-time physical activity. Example cases are based on 29 (P008) and 32 (P121) reported sessions; differing colors indicate ratings from one specific session.

Of the 48 nomothetic correlations and 48 sets (22 participants per set) of idiographic correlations, four exemplar comparisons are presented to highlight the potential for differing interpretations based on variable-centered or person-specific analyses in Fig 3. Panel A shows a case where the nomothetic result–a medium positive association between variables (motive to replenish energy; activity duration)–is reflected by a single within-person association. Panel B highlights an instance where most within-person associations reflect the direction of the nomothetic interpretation (stronger motives to be social relate to longer reported durations),

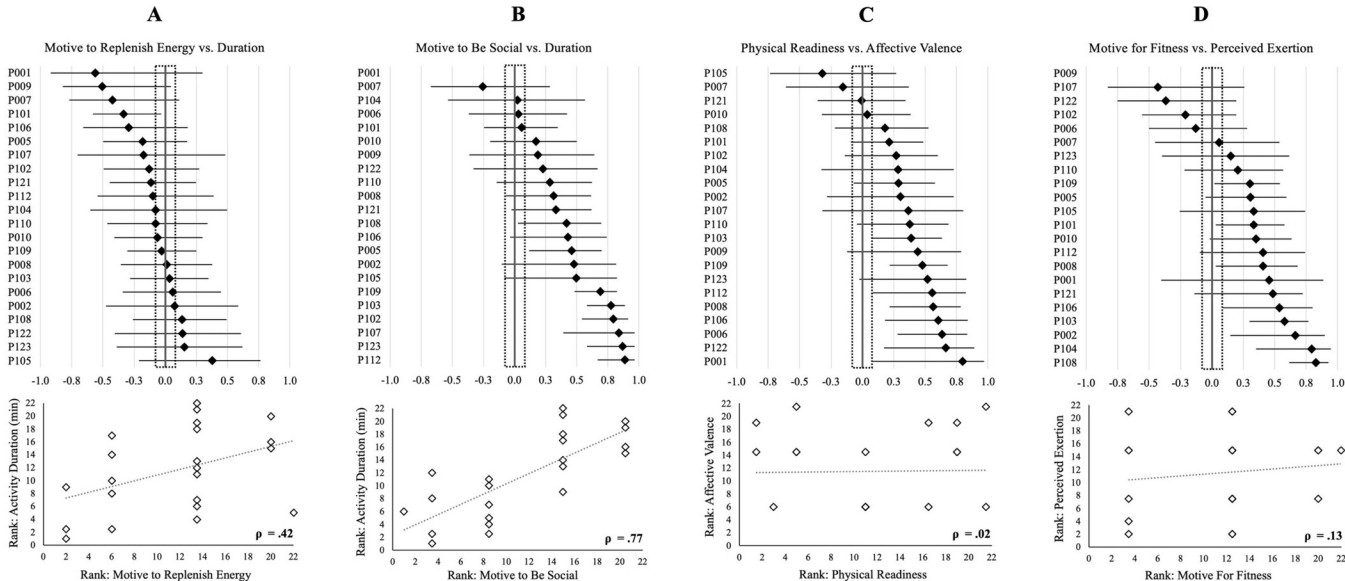

**Fig 3. Nomothetic and idiographic analyses can yield different interpretations of how readiness states and situational motivation are related to leisure-time physical activity duration, intensity, and experienced pleasure.** Exemplar cases showing direction and magnitude (Spearman's rho) of associations computed using ecological momentary assessment across 10 weeks (N = 22); forest plots show idiographic results with 95% confidence intervals (black vertical line represents 'no association'; dotted box denotes range for negligible effects) and scatter plots show nomothetic results from aggregated within-person data. Panel A: positive between-person association does not correspond with most person-specific associations. Panel B: positive between-person association corresponds with most person-specific associations. Panels C and D: negligible-to-small between-person association misrepresents rather consistent positive person-specific associations.

with varying effect sizes. Panels C and D represent cases where a negligible-to-weak nomothetic association misrepresents observations that more than half of the within-person associations are medium-to-large and in the expected direction (higher physical readiness relates to more positive ratings of recalled affective valence; stronger situational motivation for fitness relate to higher ratings of perceived exertion).

It is essential to transparently represent our whole dataset holistically to show the potential for heterogeneity in person-specific associations across all variable combinations, beyond the highlighted exemplar cases in Fig 3. Thus, Figs 4–6 contain heat maps that denote a higher versus lower number of participants categorized across 7 levels of association, with reference nomothetic associations for each variable combination. For example, while the nomothetic association suggests a medium inverse effect such that higher cognitive fatigue is related to lower LTPA duration, only two individuals demonstrated a similar within-person association (Fig 4). Remaining within-person patterns were: small (n = 5) or large (n = 1) inverse associations; negligible associations (n = 9); and small (n = 4) or medium (n = 1) positive associations. In some cases, all levels of association were represented by one or more individuals based on within-person analyses, deviating from the nomothetic interpretations (e.g., strength in motivation to organize thoughts negligibly related to perceived exertion, Fig 5; stronger motivations for health moderately related to less pleasant ratings of affective valence, Fig 6).

# Discussion

## Summary of findings

The current study was designed to explore interindividual variability in person-specific associations regarding hypothesized predictors of volitional LTPA effort and experience, as well as

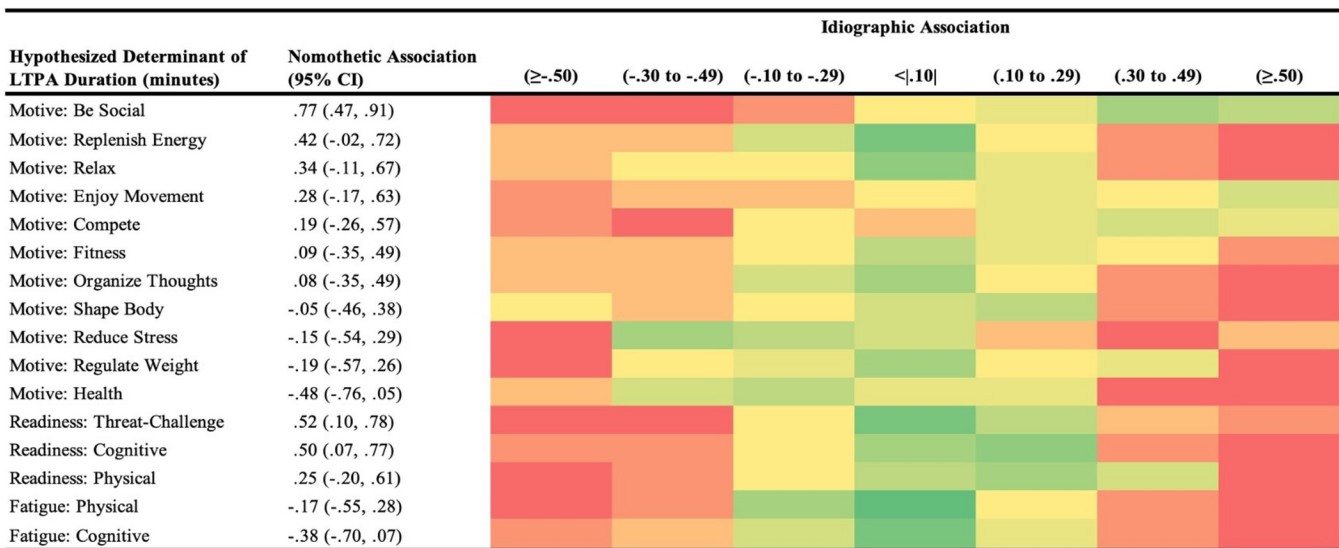

**Fig 4. Potential for heterogeneity in within-person associations across all types of situational motivation and domains of readiness with duration of leisure-time physical activity (LTPA).** Participants (N = 22) provided ecological momentary assessments across 10 weeks to indicate readiness states and motives to prior to performing LTPA; compared to nomothetic associations, the heat map denotes the number of individuals categorized into each type of association based on direction and magnitude (red = 0 individuals, darker green = higher number of individuals).

highlight the potential for differing interpretations when data are assessed across individuals. By leveraging event-contingent EMA procedures to allow participants to report on purposeful (to them) instances of LTPA, this work demonstrates that–within a relatively small sample (N = 22) over a multi-week period of time (10-weeks)–situational motivations, readiness states, LTPA behavior (session-to-session mode, duration and intensity), and affective experience can vary generally across and dynamically within individuals. Further, this exploratory data provides proof-of-concept that how researchers and practitioners interpret relationships

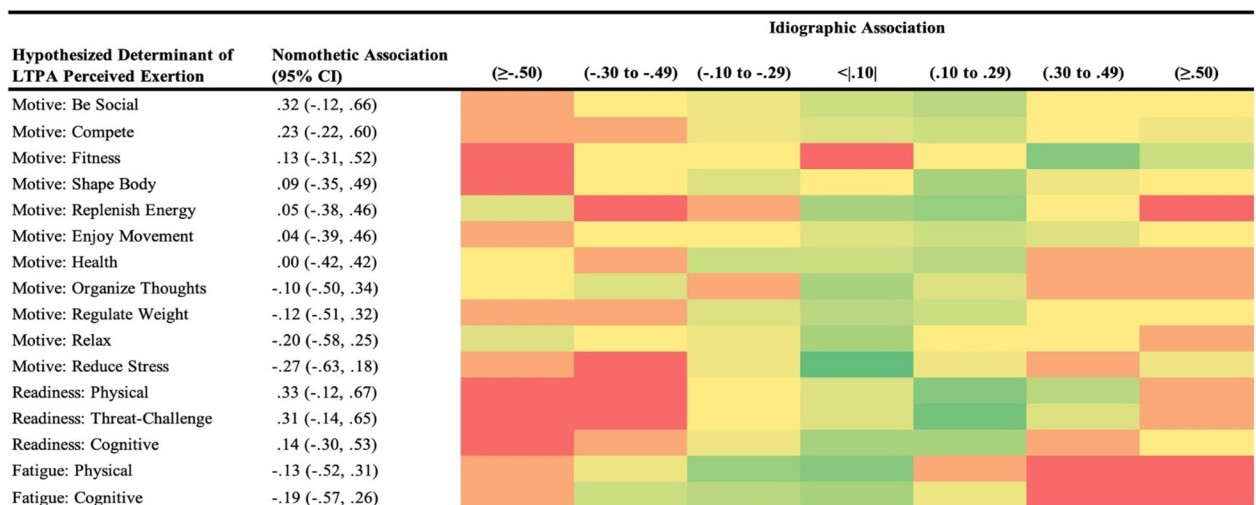

**Fig 5. Potential for heterogeneity in within-person associations across all types of situational motivation and domains of readiness with intensity of leisure-time physical activity (LTPA).** Participants (N = 22) provided ecological momentary assessments across 10 weeks to indicate readiness states and motives to prior to performing LTPA; compared to nomothetic associations, the heat map denotes the number of individuals categorized into each type of association based on direction and magnitude (red = 0 individuals, darker green = higher number of individuals).

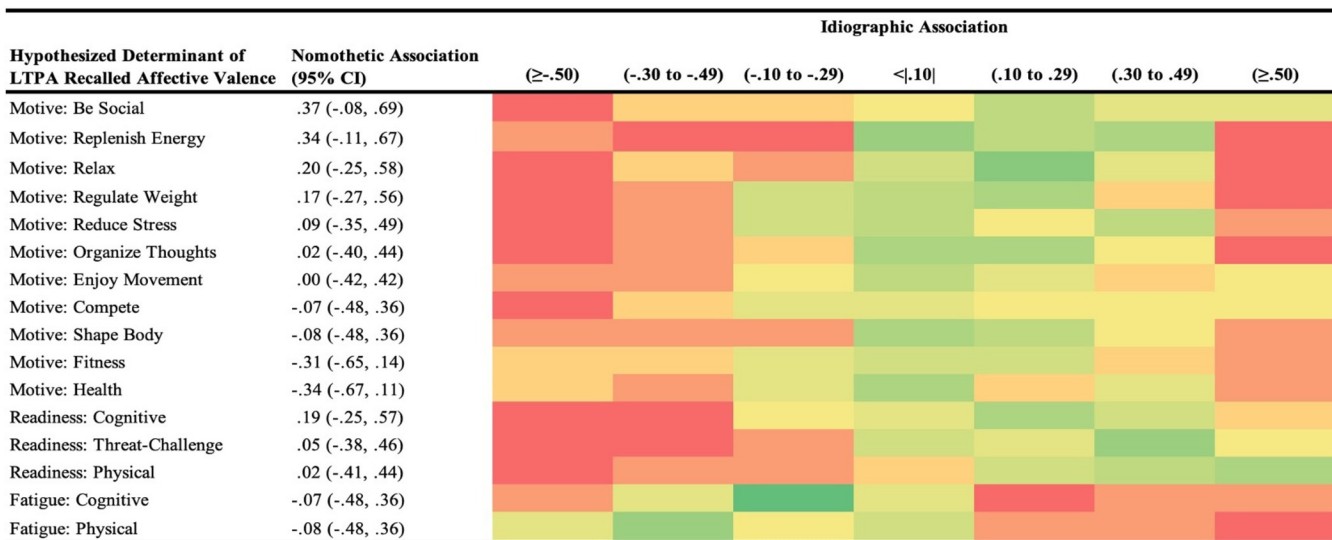

**Fig 6. Potential for heterogeneity in within-person associations across all types of situational motivation and domains of readiness with recollections of pleasure experienced during leisure-time physical activity (LTPA).** Participants (N = 22) provided ecological momentary assessments across 10 weeks to indicate readiness states and motives to prior to performing LTPA; compared to nomothetic associations, the heat map denotes the number of individuals categorized into each type of association based on direction and magnitude (red = 0 individuals, darker green = higher number of individuals).

between physical activity-related variables is likely to be impacted by the level of analysis (idiographic vs. nomothetic). This work provides the requisite foundation to stimulate future research based on both our preliminary findings and study limitations.

## Variability in reported LTPA characteristics

The observed variability across and within participants regarding LTPA in this naturalistic study reflects prior findings from single time-point or shorter term studies designed to quantify the complexity of physical activity behavior. Prior data collected for cross-sectional [49], as well as 7-day [50] and 14-day [51, 52] EMA designs demonstrate that sample populations report engaging various primary modes relating to aerobic, resistance, combined aerobic and resistance (e.g., Tae Bo, CrossFit), and group sports in their leisure time, in line with data provided in Table 2. Low to moderate consistency in session-by-session intensity (objectively measured and perceived) and volume (intensity multiplied by duration) has also been documented for volitional LTPA [50].

Research has also demonstrated that experiential responses vary across individuals [38, 40], as well as within individuals over several imposed exercise sessions [53] or over 7 to 28 days [50, 54–56]. Such variability reflects the expectation that physical activity-related experiences are multifaceted (physical, psychological, social effects), which may differ across mode and context [57]. Resultant experiences from physical stimuli are likely to be influenced, in part, by pre-activity perceptions of readiness and multi-thematic situational motivations, which have been shown to fluctuate over time [50, 58]. As demonstrated in this study (Fig 2), the degree of fluctuation may also be relatively person-specific, warranting further research to understand how low-to-high variance explains or predicts individuals' physical activity behavior and related experiences.

## Level of analysis and differing data interpretation

Our findings present the possibility of Simpson's paradox regarding relationships between situational motivation, readiness, and target physical activity outcomes. Simpson's paradox refers to instances where an association at the population level may be reversed within the subgroups comprising that population [59]. Using simulation models, Molenaar [60] suggests that–mathematically–data derived from interindividual variation cannot be generalized to results based on intraindividual variation. In acknowledging this possibility, Kievit [13] cautioned that differences in interpretation, especially when the direction of association are in opposition, can have significant implications for delivery of care in medical and social contexts. We offer two scenarios to demonstrate how differing interpretations may unfavorably impact practice related to physical activity promotion.

From our data, we consider the observed nomothetic pattern that individuals with stronger general motives to replenish energy tended to perform longer LTPA, on average. If this relationship is uncritically assumed to exist within individuals, an exercise professional may urge clients with relatively low energy levels to engage in longer-than-normal bouts of activity, reflecting the traditionally held belief that 'exercise makes people feel better' [40]. However, this interpretation opposes our preliminary observation that the larger proportion of participants demonstrated an inverse association between variables, meaning that stronger motives for replenishing energy were associated with lower reported duration over time.

The current analyses also revealed that the nomothetic interpretation that physical readiness had almost no association with recalled affective valence. This lack of association opposes a growing body of qualitative findings, which identify factors underlying perceptions of physical readiness (e.g., energy, fitness, body integrity) as influencing physical activity-related cognitions and affective experiences [61–63]. Relying solely on the nomothetic interpretation, a client's poor perception of physical readiness may not dissuade a practitioner from imposing a scheduled, more strenuous exercise session–potentially eliciting a rather unpleasant experience. Conversely, based on observations that individuals reported LTPA sessions that were preceded by more unfavorable readiness states, it would also be shortsighted to conclude that physical activity should be generally be avoided completely if physical, mental, and emotional states are suboptimal.

## Future directions in data analysis and intervention development

The potential for emergent subgroups, based on direction and magnitude of idiographic associations, has implications for future statistical analyses. The process of first examining person-specific data is recommended prior to estimating within-person and between-person effects using repeated-measures correlation or subsequent multi-level modeling [32], as the observation of small effects could be due to consistently small effects across subjects (i.e., good model fit) or an artifact of heterogeneous slopes (i.e., poor model fit). Sufficient time-series data can also be leveraged to conduct network analyses [21]. For example, group iterative multiple model estimation (GIMME) procedures can be applied to create personalized maps comprised of nodes (variables operationalizing target constructs) and edges (indicating relation between nodes that are unique to each individual or common across the sample) [64, 65]. Given that GIMME can adequately map between 3 and 20 variables, this approach may be particularly suitable to understand which types of situational motivation and domains of readiness are most powerful–for whom and in general–for predicting physical activity-related effort and experience. In line with an advancing focus on precision health [66], the need for robust idiographic exploration (and later replication and confirmation of findings) extends beyond our

selected variables and pertains broadly to hypothesized determinants of physical activity adoption and adherence [10, 11] to effectively inform person-adaptive treatment approaches.

In anticipation of applying network approaches to broadly understand and quantify determinants of physical activity performance and experience (including our selected variables of situational motivation and readiness), we propose several key considerations. Given the potential for numerous different types, contexts and experiences of physical activity, the first consideration is to narrow the focus of subsequent studies. For instance, specifically assessing the impact of situational motivation and readiness on perceived pleasure during cycling exercise in cardiac rehabilitation may yield more meaningful, directly translatable information for that context more so than analyzing various forms for physical activity and assuming inferences are broadly transferable to all physical activity-related situations. The second consideration is that longer (or repeated) observation periods may be required. While a generalized target of 60 data points per person [67] has been recommended for GIMME, simulation studies should also be conducted to tailor sample size needs to specific research questions [21]. For example, Dai et al. [68] produced person-specific models for physical activity (in addition to other variables, such as mental sharpness and social engagement) with 36 participants who had 13–25 data points. As repeated performance of physical activity elicits physiological and psychological adaptations, it is reasonable to consider that initial person-specific maps of determinants may evolve (or become obsolete) over time. Thus, our third consideration is for future research to assess the utility of creating person-specific maps based on EMA of forecasted responses to hypothetical physical activity sessions, in order to collect the requisite amount of data in a more timely manner, without repeated physical stimuli.

It is critical to understand which determinants of LTPA effort and experience are specific to an individual or generalizable across groups to develop, refine, and tailor behavioral interventions. For example, developing hypothesized pathways by which a treatment can support physical activity from both nomothetic and idiographic perspectives aligns with calls for applying flexible, iterative optimization processes to candidate treatments prior to conducting randomized controlled trials [69]. Accounting for intraindividual variation and integrating person-specific determinants stands to improve recipients' acceptability of a physical activity intervention, thusly enhancing engagement, compliance to the program, and long-term behavioral adherence [70].

Optimal intervention development and research also necessitates sufficient and replicable descriptions of active components. Given that enactment of FNLP–the exercise programming model that informed study variables–is hypothesized to foster more self-determined forms of motivation [28], intervention development efforts should incorporate terms and definitions according to the taxonomy of motivation and behavior-change techniques proposed by Teixeira and colleagues (2020) for health promotion contexts [71]. For example, FNLP 'provides choice' and encourages self-experimentation [28] as *autonomy-support techniques* to 'assist in setting optimal challenge' based on situational motivation and readiness states as a *competence-support technique*. As FNLP may be implemented by a professional (e.g., certified exercise physiologist, behavioral interventionist) who educates patients or clients, a complementary taxonomy proposed by Ahmadi and colleagues (2023) will also benefit intervention development and assessment efforts [72]. This taxonomy importantly specifies behaviors *of the educator* that can support or thwart autonomous motivation, in recognition outcomes are differently impacted via separate pathways (e.g., psychological need satisfaction and need frustration) [73].

## Limitations

While we prefaced this work as exploratory and caution against premature inferences based solely our findings, we also acknowledge several limitations. First, these data were collected using a convenience sample of individuals who self-selected into the study, who can be described as rather active with high competences in support of a physically active lifestyle. For example, the overall perceived physical fitness and physical activity-related competency scores in the current sample were higher compared to prior reports [43, 74, 75]. Further research must be conducted to understand within-person associations in varied sample populations. A second limitation is that results based on event-contingent reporting of volitional LTPA may be skewed more positive, as individuals could be less likely to initiate reports in response to strongly negative experiences. Additionally, despite the prompts specifically asking individuals to recall feelings during activity, capturing data in the post-activity stay may still yield higher-than-actual ratings of pleasure. While this approach had been used previously [61] to over-come limitations in collecting in-task data, it will be necessary to replicate these methods in intervention or treatment settings, paired with wearable devices or direct observation to more accurately monitor physical stimuli and implement approaches to capture actual in-task ratings of affect.

This work is also limited by potential influences of translating survey items; the BMZI and PAHCO were translated from German to English and the ARMS, BREQ-3, and PPFS were translated from English to German. While structured forward-backward translation procedures (including bi-lingual native speakers) were applied, further validation is needed for surveys targeting hypothesized determinants and outcome variables. Specifically, the predictive validity of single-item measures is imperative [76], as their use has practical importance for minimizing survey fatigue [77]. A final limitation relates to the high degree of variability in the number of sessions reported; less data can compromise the quality of correlation analyses for certain individuals. Conversely, while participants in the current sample were explicitly asked to maintain normal patterns of behavior and were not given a minimum number of sessions to perform or incentivized to initiate EMA reports, we cannot with certainty rule out the possibility for reactivity. It is important to acknowledge that perceived enforcement of behavior (e.g., performing additional or undesired LTPA due to being observed by researchers) would compromise the aim to understand volitional behavior and experiential aspects thereof. In future studies, it may be beneficial to conduct debriefing assessments (quantitative surveys or qualitative interviews) to understand the potential for reactivity and determine strategies to minimize such effects. Thus, while the 10-week study described in this paper is substantially longer that most existing single-wave EMA studies pertaining to physical activity, which generally aim to capture data across several days or several weeks [78–80], we re-iterate the necessity of subsequent research to appropriately estimate person-specific or person-oriented effects with minimal reporting bias.

## Conclusions

The findings presented in the current paper importantly demonstrate the potential for heterogeneity regarding how hypothesized determinants (situational motivation, readiness) are associated with effort- and affect-related outcomes of LTPA. The ability to predict, and thusly manage, physical activity dosage and experience is a high priority, as these impact subsequent physiological adaptations (necessary for reducing disease risk and severity) and behavioral repetition (necessary to sustain any acquired benefits). Continued research is warranted to understand dynamic psychological processes underlying physical activity behavior. Uncovering personalized models that identify LTPA determinants represents a key step to realize goals for

precision behavioral medicine applied to physical activity and exercise in support of long-term health and well-being.

## Supporting information

**S1 Checklist. Global inclusivity in research questionnaire.**
(DOCX)

**S1 Data. Open access data file.**
(XLSX)

## Acknowledgments

The authors would like to acknowledge the efforts of Michael Zweier and Oliver Neumann in facilitating participant recruitment and data collection.

## Author Contributions

**Conceptualization:** Kelley Strohacker, Gorden Sudeck, Adam H. Ibrahim, Richard Keegan.

**Data curation:** Kelley Strohacker, Gorden Sudeck, Adam H. Ibrahim.

**Formal analysis:** Kelley Strohacker.

**Investigation:** Kelley Strohacker.

**Methodology:** Kelley Strohacker, Gorden Sudeck, Richard Keegan.

**Project administration:** Kelley Strohacker, Gorden Sudeck, Adam H. Ibrahim.

**Resources:** Kelley Strohacker, Gorden Sudeck.

**Software:** Kelley Strohacker, Gorden Sudeck.

**Supervision:** Kelley Strohacker, Gorden Sudeck.

**Visualization:** Kelley Strohacker.

**Writing – original draft:** Kelley Strohacker, Gorden Sudeck, Adam H. Ibrahim, Richard Keegan.

**Writing – review & editing:** Kelley Strohacker, Gorden Sudeck, Adam H. Ibrahim, Richard Keegan.

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
