## [Decision Letter · Decision Letter 0]

5 Apr 2024

PONE-D-24-08085Exploring heterogeneity in person-specific associations of situational motivation and readiness with leisure-time physical activity duration, intensity, and affective valencePLOS ONE

Dear Dr. Strohacker,

Thank you for submitting your manuscript to PLOS ONE. After careful consideration, we feel that it has merit but does not fully meet PLOS ONE’s publication criteria as it currently stands. Therefore, we invite you to submit a revised version of the manuscript that addresses the points raised during the review process.

We look forward to receiving your revised manuscript.

Kind regards,

Zulkarnain Jaafar

Academic Editor

PLOS ONE

Journal Requirements:

2. Please include a complete copy of PLOS’ questionnaire on inclusivity in global research in your revised manuscript. Our policy for research in this area aims to improve transparency in the reporting of research performed outside of researchers’ own country or community. The policy applies to researchers who have travelled to a different country to conduct research, research with Indigenous populations or their lands, and research on cultural artefacts. The questionnaire can also be requested at the journal’s discretion for any other submissions, even if these conditions are not met.  

Please find more information on the policy and a link to download a blank copy of the questionnaire here: https://journals.plos.org/plosone/s/best-practices-in-research-reporting. 

Please upload a completed version of your questionnaire as Supporting Information when you resubmit your manuscript.

**Additional Editor Comments:**

Please attend to all comments provided by the reviewers and make the necessary corrections or changes.

Reviewers' comments:

Reviewer's Responses to Questions

**Comments to the Author**

1. Is the manuscript technically sound, and do the data support the conclusions?

Reviewer #1: Yes

Reviewer #2: Partly

2. Has the statistical analysis been performed appropriately and rigorously? 

Reviewer #1: Yes

Reviewer #2: Yes

3. Have the authors made all data underlying the findings in their manuscript fully available?

Reviewer #1: Yes

Reviewer #2: Yes

4. Is the manuscript presented in an intelligible fashion and written in standard English?

Reviewer #1: Yes

Reviewer #2: Yes

5. Review Comments to the Author

Reviewer #1: I would like to thank for the opportunity to review this highly interesting manuscript. Overall, this manuscript is well written, and it presents important topic of motivation and physical activity. I have only some comments to further improve the quality of this manuscript.

Title

Overall, the title of this manuscript is an accurate presentation of this manuscript content. However, perhaps Authors could shorten their title in terms of clarity.

Abstract

The abstract of this manuscript provides the most important information for the reader. However, one issue of this manuscript is that in some cases Authors use very long sentences that might be hard to follow for the reader. For example, the last sentence of the abstract is written on five lines.

Main text

Although the introduction of this manuscript is well written, some concepts are not introduced, and some findings are presented in too much detail. Authors are recommended to provide more general introduction of previous findings and avoid too many details about specific previous research papers. Authors are using scales that measure different forms of motivation from the self-determination theory, but in the introduction, there is no information at all about the self-determination theory.

Methods section is a bit messy. For example, there is information under the study design that should be under the data analysis section. Participants are poorly described and there are several paragraphs about participants (could be only one). Overall, the methods could be organized better.

Overall, the results and described in a great detail.

Tables are formatted in different style.

The quality of figures could be improved.

Overall, the discussion of this manuscript is well written. However, the discussion could be structured better. Also, more suggestions for future research could be provided. Specific recommendations for future motivational interventions could be given with the aim to increase physical activity behavior. There are several recent high level classification systems of motivational behaviors recommended in self-determination theory interventions that could be used to design need-supportive motivational intervention with the aim to increase physical activity behavior.

Reviewer #2: The manuscript titled "Perceptual responses to high- and moderate-intensity interval exercise in adolescents" Exploring heterogeneity in person-specific associations of situational motivation and readiness with leisure-time physical activity duration, intensity, and affective valence’. Overall, the manuscript is very well-written and clearly describes the reason for the research, what gap the research is filling, and accurately concludes with the impact of the results on the field. The research question and results would be of interest to the readers of the journal. My enthusiasm for the paper remains despite the following issues that need to be addressed as highlighted below:

Is there any specific reason why the researcher chose 10 weeks for conducting the observation? Since the participants who enrolled in this study underwent multiple procedures (e.g. answering the questionnaire pre-post activity) across the 10 weeks, this situation can lead to an enforcement type of situation rather than a volitional-based PA situation. Consequently, most probably the participants come out with any kind of activity just to complete the task. How did the researcher control this possible situation? If this happens, the intended volitional LTPA effort and affective experiences will be compromised. Any comments?

Line 138: why a minimum of 60 minutes per week of MVPA was chosen as one of the intrinsic criteria of the participants? Any specific reason?

Line 144-148: it would be useful to include the sample size calculation to provide the readers with the minimum number of participants required for this study. What indicator that the researcher uses to decide whether the number of participants was sufficient for the analysis based on the snowball sampling technique?

Line 227 – 232: Evidence has shown that post-affective responses/experience did not reflect on individual exercise/PA adherence and no such thing as recalled in-task affective responses. Affective responses measured via a single-item scale FS usually conducted during the exercise/PA, not after the completion of the exercise/PA. This is because post-exercise feelings will lead to homogeneous pleasurable feelings regardless of the type and intensity of the activity. Any comments?

6. PLOS authors have the option to publish the peer review history of their article (what does this mean?). If published, this will include your full peer review and any attached files.

Reviewer #1: No

Reviewer #2: **Yes: **Adam Abdul Malik

---

## [Author Response · Author response to Decision Letter 0]

20 May 2024

We have provided a document within the submission that addresses all reviewer comments, using point-by-point responses.

---

## [Decision Letter · Decision Letter 1]

19 Jun 2024

PONE-D-24-08085R1Exploring person-specific associations of situational motivation and readiness with leisure-time physical activity effort and experiencePLOS ONE

Dear Dr. Strohacker,

Thank you for submitting your manuscript to PLOS ONE. After careful consideration, we feel that it has merit but does not fully meet PLOS ONE’s publication criteria as it currently stands. Therefore, we invite you to submit a revised version of the manuscript that addresses the points raised during the review process.

**ACADEMIC EDITOR: Dear Author, Please consider the suggestion made by the reviewer/s. Thanks.** The decision of this manuscript is justified based on PLOS ONE’s publication criteria and not on its novelty or perceived impact.

We look forward to receiving your revised manuscript.

Kind regards,

Zulkarnain Jaafar

Academic Editor

PLOS ONE

Journal Requirements:

Reviewers' comments:

Reviewer's Responses to Questions

**Comments to the Author**

1. If the authors have adequately addressed your comments raised in a previous round of review and you feel that this manuscript is now acceptable for publication, you may indicate that here to bypass the “Comments to the Author” section, enter your conflict of interest statement in the “Confidential to Editor” section, and submit your "Accept" recommendation.

Reviewer #1: All comments have been addressed

Reviewer #3: (No Response)

2. Is the manuscript technically sound, and do the data support the conclusions?

Reviewer #1: Yes

Reviewer #3: Yes

3. Has the statistical analysis been performed appropriately and rigorously? 

Reviewer #1: Yes

Reviewer #3: Yes

4. Have the authors made all data underlying the findings in their manuscript fully available?

Reviewer #1: Yes

Reviewer #3: Yes

5. Is the manuscript presented in an intelligible fashion and written in standard English?

Reviewer #1: Yes

Reviewer #3: Yes

6. Review Comments to the Author

Reviewer #1: Overall, Authors have done well job on revising their manuscript. I have only one comment. Specifically, Authors have argued that intervention development efforts should incorporate terms and definitions according to the taxonomy of motivation and behavior-change techniques proposed by Teixeira and colleagues (2020) for health promotion contexts. I would like to add that for educational context, a classification system of motivational behaviors by Ahmadi and colleagues (2023) was recently developed. Health-related messages are often provided to children in the educational context (e.g., during physical education classes). The advantage of Ahmadi and colleagues (2023) classification system of motivational behaviors is that it also lists need-thwarting behaviors. It is important to not only provide need support, but also to avoid need thwarting, because need satisfaction and need thwarting are related to adaptive and maladaptive outcomes via separate pathways (Haerens et al., 2015). Thus, it is important to adopt need-supportive behaviors and at the same time avoid need-thwarting behaviors. There are also more advantages of Ahmadi and colleagues (2023) classification system of motivational behaviors such as more experts were involved in Ahmadi and colleagues (2023) Delphi panel compared to Teixeira and colleagues (2020) study (34 vs 18 experts).

Ahmadi, A., Noetel, M., Parker, P., Ryan, R. M., Ntoumanis, N., Reeve, J., Beauchamp, M., Dicke, T., Yeung, A., Ahmadi, M., Bartholomew, K., Chiu, T. K. F., Curran, T., Erturan, G., Flunger, B., Frederick, C., Froiland, J. M., González-Cutre, D., Haerens, L., . . . Lonsdale, C. (2023). A classification system for teachers’ motivational behaviors recommended in self-determination theory interventions. Journal of Educational Psychology, 115(8), 1158–1176. https://doi.org/10.1037/edu0000783

Haerens, L., Aelterman, N., Vansteenkiste, M., Soenens, B., & Van Petegem, S. (2015). Do perceived autonomy-supportive and controlling teaching relate to physical education students’ motivational experiences through unique pathways? Distinguishing between the bright and dark side of motivation. Psychology of Sport and Exercise, 16, 26–36. https://doi.org/10.1016/j.psychsport.2014.08.013

Reviewer #3: Comments to Author

Overall

This study investigates the association between psychological constructs (e.g., situational motivation and readiness) and leisure-time physical activity (PA). Specifically, that authors explore how idiographic analytical approaches might be used when EMA data is available to assess individualized associations between psychological constructs and PA. The authors note differences between idiographic and nomothetic approaches and discuss how idiographic approaches may be useful for patient engagement and tailoring of interventions. This work is a good case study for behavioral interventionists interested in utilizing technology and tailoring interventions on the potential use of idiographic methods.

The authors responded sufficiently to previous reviewer comments and appropriately discuss the limitations of this work and its implications for future research.

Abstract

The abstract is clear.

Introduction

The motivation for the work is described in a logical sequence and the author’s framing of the work as a proof of concept appears acceptable.

Methods

The instruction given to participants are clearly described. The sample size justification is appropriate for a study of this type. All measures are discussed with sufficient references.

Results

The results are sufficiently explained.

Discussion

References provided in the discussion are helpful to readers who are interested in further applying these idiographic methods.

Tables/Figure

Figures are of publishable quality. Legends are provided as appropriate.

7. PLOS authors have the option to publish the peer review history of their article (what does this mean?). If published, this will include your full peer review and any attached files.

Reviewer #1: No

Reviewer #3: No

---

## [Author Response · Author response to Decision Letter 1]

2 Jul 2024

We have provided written responses in a separate documented labeled 'Response to Reviewer Comments'

---

## [Editor Report · Decision Letter 2]

4 Jul 2024

Exploring person-specific associations of situational motivation and readiness with leisure-time physical activity effort and experience

PONE-D-24-08085R2

Dear Dr. Strohacker,

We’re pleased to inform you that your manuscript has been judged scientifically suitable for publication and will be formally accepted for publication once it meets all outstanding technical requirements.

Kind regards,

Zulkarnain Jaafar

Academic Editor

PLOS ONE
---

## [Editor Report · Acceptance letter]

10 Jul 2024

PONE-D-24-08085R2 

PLOS ONE

Dear Dr. Strohacker, 

I'm pleased to inform you that your manuscript has been deemed suitable for publication in PLOS ONE. Congratulations! Your manuscript is now being handed over to our production team.

Kind regards, 

on behalf of

Dr. Zulkarnain Jaafar 

Academic Editor

PLOS ONE